# Electric Discharge Machining on Stainless Steel Using a Blend of Copper and Fly Ash as the Electrode Material

**DOI:** 10.3390/ma15196735

**Published:** 2022-09-28

**Authors:** Ponnambalam Balamurugan, Marimuthu Uthayakumar, Manickaraj Pethuraj, Dariusz Mierzwiński, Kinga Korniejenko, Mohd Shukry Abdul Majid

**Affiliations:** 1Faculty of Mechanical Engineering, Kalasalingam Academy of Research and Education, Krishnankoil 626126, India; 2Faculty of Mechanical Engineering and Technology, University Malaysia Perlis (UniMAP), Arau 02600, Perlis, Malaysia; 3Department of Mechanical Engineering, Saveetha School of Engineering, Saveetha Institute of Medical and Technical Sciences, Tamil Nadu 602117, India; 4Faculty of Material Engineering and Physics, Cracow University of Technology, Jana Pawła II 37, 31-864 Cracow, Poland

**Keywords:** electric discharge machining (EDM), copper, fly ash, tool wear

## Abstract

In the current work, several composites made with fly ash reinforcements are used to conduct electrical discharge machining (EDM) on stainless steel that is commercially accessible. Four composites were prepared with 2.5 to 10% reinforcement of fly ash with steps of 2.5%, copper is used as the matrix material. The specimens were created using the powder metallurgy method, which involved compaction pressures of 450 MPa and 900 °C for 90 min of sintering. The prepared composites are used as the electrode tool for EDM. EDM studies were carried out at two different current amplitudes (5A and 15A) by maintaining the Pulse on time (100 µs), Pulse off time (50 µs), and the depth of machining as 2 mm. The findings show that the addition of more fly ash to the copper matrix increased the material removal rate when cutting the SS304 plate and had a negative impact on the tool. The composite loses its ability to transfer heat during machining as the level of fly ash increases, raising the temperature in the copper matrix and causing the copper to melt more quickly at the electrode interface during machining, leading to increased electrode wear. While tool life was reduced because of the increase in current amplitude, machinability was enhanced.

## 1. Introduction

The demand for high-strength materials is continuously increasing in the engineering industries due to advancement in the field of materials and theability of machines to make them harder [1,2]. Electric discharge machining (EDM) is one of the most preferred methods for producing parts of complicated profiles with high accuracy. The machining of hard and brittle materials is preferred through EDM, as machining in conventional processes is expensive and incurs a high tool wear rate [3,4,5]. Discrete electrical sparks produced by electrical energy are utilized in the EDM machining process to erode material away from the workpiece [6,7]. The workpiece and the tool are submerged in the dielectric medium during this operation. The workpiece and the tool are submerged in dielectric fluid while being separated by a very small gap during EDM cutting. The material on the workpiece’s surface reaches extreme temperatures due to the discharge energy during the spark, which causes the material to burn [8,9]. The residue is then flushed away by the dielectric medium. The various parameters that affect the results of EDM are the pulse current, servo voltage, pulse on time, pulse off time, dielectric medium, and electrode material [10,11,12].

Numerous studies investigated the impact of various electrode materials on several variables, including the rate of material removal, electrode wear, and overcut [13,14,15]. Based on input parameters such as voltage, pulse current, and pulse-on-time, Phan et al. [16] researched the EDM of SKD61 die steel and discovered the ideal level for maximal MRR, minimal surface roughness, and decreased white layer thickness. When Genç et al. [17] investigated boron alloy steels, they discovered that both MRR and surface roughness increase as the discharge current and the duration of the stroking increase. According to research by Korlos et al. [18], who examined the delamination effect on CFRP laminates under various pulse-on-time conditions, the delamination factor increases as the pulse-on-time increases, in contrast to the MRR decreasing since the melted material is not completely flushed out by the dielectric fluid, which causes delamination. Urtekin et al. [19] investigated the wire EDM of biodegradable AZ91 Mg alloy and discovered that with high power level setting, i.e., with the higher pulse on time and lower pulse off time, higher surface roughness is formed on the machined surface as a result of the formation of deeper and wider craters due to high energy discharges in the machining gap. Several attempts have been made to find suitable composite material for the electrode to increase the performance during machining on an electrical discharge machine [20,21,22,23,24]. Preparation of composite electrodes by powder metallurgy technique is preferred by the researchers since it is not feasible to prepare the composites with high variation in densities by other methods such as stir casting, since during the melting, due to high variation in densities, either the reinforcement floats or it settles at the bottom before pouring into the mold [25,26]. Powder metallurgy is the process by which components are made by blending the powder and pressing them in the die to make them mechanically bonded together to give the required shape and then heating them to the required temperature to make them metallurgically bonded. Alekhya et al. [27] investigated the properties of Al-TiO_2_-Mg composites and discovered that the specimen’s density increases with increasing sintering temperature, but the composite’s hardness decreases with higher sintering temperatures.

In the present study, a novel composite material is prepared using the powder metallurgy technique and is used as the electrode material to machine stainless steel, and the effects of the influence of reinforcement on parameters such as the rate of material removal and wear on the electrode were studied. However, some previous research was carried out in this topic; the literature still shows the research gap of including improvements in efficiency, cost-effectiveness, and environmental aspects for this technology [28,29]. The mixture of copper and fly ash as an electrode material is researched. The previous research confirms possibility of the application of this kind of composite for similar purpose [30,31]. Furthermore, the addition of fly ash makes technology more environmentally friendly due to the use of anthropogenic waste from the energy industry [5,32].

## 2. Materials and Methods

Powder metallurgy was used to create the electrode material, which is made up of fly ash as a reinforcing material and copper as a matrix in various ratios (0, 2.5, 5, 7.5, and 10%). The matrix was made of copper metal powder, which was acquired from Oxford Chemicals and had a purity level of 99.7% with an average particle size of 30 μm. It is an extremely high thermal and electrical conductivity metal that is soft, malleable, and ductile. The copper had a melting point of 1085 °C and a density of 8940 kg/m^3^.

One of the byproducts of combustion, fly ash is a tiny particle that rises with flue gases as flue particles. The ash created when coal is burned is commonly referred to as “fly ash.” Fly ash has a density of 890 kg/m^3^ and a melting temperature of 1400 °C. Fly ash has low density, strong wear, and abrasion resistance qualities. An average particle size of 5 μm is used for the fly ash reinforcement, which is sourced from the Tuticorin Thermal Power Station in India. Fly ash delivered had the following composition. SiO_2_ makes up 61.75 percent of the material, followed by Al_2_O_3_ (25.24%), Fe_2_O_3_ + Fe_3_O_4_ (4.77%), CaO (1.3%), and MgO (0.87%).

The SEM (MAKE: CARLZEISS, MODEL: EVO18) investigation was performed for both raw materials. Figure 1a,b demonstrate the morphology of copper and fly ash, respectively, obtained by a scanning electron microscope. According to the SEM pictures, used fly ash was spherical and copper powder had a dendritic structure. From the SEM micrograph the copper particles of varying sizes up to 50 µm and the fly ash particles are approximately 1 µm because of the smaller particle size of the fly ash (reinforcement) the possibility of agglomeration is more for reinforcement due to its adhesive nature. 

By weighing the powders with a digital balance (MAKE: SHIMADZU, MODEL: ATX224) that has a 0.0001 g accuracy, the copper and fly ash powders are measured in the proper weight ratios. In order to reduce the moisture content before compacting, the taken proportions are manually blended and heated to 100 °C in an argon environment. In the Universal Testing Machine (UTM), compaction was carried out using a single-action compaction die at a compaction pressure of 450 MPa after pre-heating. The interior surfaces of the die are cleaned with acetone prior to the compaction process, and after drying, the surfaces are lubricated with wax to prevent the powder from adhering to the die and to facilitate the smooth removal of the specimen from the die. Green compact is the term referring to the product created during the compaction process (raw material). The diameter of the green compact specimen after preparation was 20 mm. Only mechanical bonds hold the particles together in a green compact. The green compact was sintered in a tube furnace for 90 min at 900 °C in an argon environment to metallurgically bond the particles. For additional testing, the sintered specimen was machined to the necessary dimension.

Table 1 displays the green density and sintered density of produced samples.

EDM was carried out using five different types of electrodes which were prepared through powder metallurgy process. Tool (Electrode) material is composite prepared with copper as matrix and varying proportions of fly ash from 0 to 10% in steps of 2.5% to conduct EDM studies on the commercially available SS304 plate which was used as workpiece. The chemical composition of SS304 is as follows: Carbon (0.07%), Chromium (17.5% to 19.5%), Manganese (2%), Silicon (1%), Phosphorus (0.045%), Sulphur (0.015%), Nickel (8–10.5%) and Iron (Remaining %). The machining parameters used in EDM machine are shown in Table 2. The experimental setup is shown in Figure 2 with EDM Machine in Figure 2a and tool with workpiece in Figure 2b. The machining was carried out by keeping Pulse on time, Pulse off time, and machining depth constant. The experiments were carried out with two different current amplitudes, as shown in Table 2.

Machining was carried out for 2 mm depth and maintained constant to find the influence of current amplitude and the electrode material. The electrode’s and the workpiece’s polarities were kept negative and positive, respectively. The electrode used for the machining was 20 mm in diameter. After machining, the workpiece and the electrode were cleaned with a dry cotton cloth. A digital weighing balance with an accuracy of 0.0001 g was used to measure the mass of the workpiece before and after machining to calculate the amount of material removed. When the ratio between the mass of the removed material and its density is calculated, the volume of the substance is determined. The ratio between the amount of material removed and the machining time was used to compute the material removal rate (MRR). Calculating tool wear involves determining the mass difference in the electrode prior to and after machining. The surface roughness of the machined surface is measured using a surface roughness tester (Make: Mitutoyo, Model: SJ40), which is a contact type surface roughness tester in which the probe is allowed to move over the machined surface by maintaining contact. The measurement distance is maintained as 3 mm over which the probe is moved on the machined surface. Two measurements of average surface roughness (Ra) were taken, and the average of two measurements is taken for plotting the graph.

## 3. Results and Discussion

The picture below displays the SEM micrographs of several composites created using the P/M approach, as well as pure copper. Figure 3 shows the matrix (copper) as a white area and the reinforcement as a black area (fly ash). The agglomeration of fly ash in the composite rises as the reinforcement ratio does, as shown in Figure 3b–e. It is evident from Figure 3c,e which represent the composites with 5% and 10% reinforcement, in Figure 3c, that the agglomeration of fly ash is less compared to the micrograph shown in Figure 3e.

Figure 4 displays the EDX mapping of various composite elements. Through Figure 4a,b, it is evident that the white background in the SEM images is copper. It is clear from Figure 4c–f that the fly ash is represented by the black areas in the SEM picture (Figure 4a) represents the components Si, Al, O, and Fe. The black background in Figure 4a represents the fly ash which can be seen from Figure 4b since in the same area copper is not available in the same area, which shows that the area represents the reinforcement. The same is proved from the Figure 4c–e which shows the higher concentration of Si, Al and O which contributes to the major composition of fly ash. In Figure 4f the Fe concentration is sparse due to less availability of Fe in the fly ash according to the chemical composition.

Figure 5 shows the results of the MRR of the SS304. The SS 304 is machined with five different electrodes under 5A and 15A current with other parameters kept constant which is shown in Table 2. From the results, it is inferred that as the ampere A increases from 5 to 15, irrespective of the electrode used, the MRR increases. However, for the same amperage used, the composite electrode with 10% fly ash shows a better MRR compared to the other electrodes. As the composition of the fly ash increases, the MRR is also increased for both 5A and 15A conditions, and a similar trend is noted on the tool wear.

Figure 6 shows that, under both ampere circumstances, tool wear increases as the fly ash content in the tool increases. As the ampere increases from 5A to 15A there is a drastic increase in the tool wear rate. The slope of the tool wear rate increases with an increase in fly ash content.

Figure 7 shows the roughness of the surface. Since the temperature of the arc increases with an increase in current amplitude, more material is melted or vaporized with each pulse, resulting in an increase in surface roughness of the machined surface. The surface roughness of the machined surface increases as the percentage of reinforcement in the tool material increases. This is because as the reinforcement percentage increases, the agglomeration in the electrode increases, which causes material to be lost in significant amounts during machining. Regardless of the rise in current amplitude, the trend for the change in surface roughness is the same.

Taking into account the broader context, it is worth noting that tool wear increases with the addition of fly ash. It makes the technology more environmentally friendly and cost-effective [5,33]. Noting that the cost of parts manufactured by the EDM methods is strongly dependent on the tooling cost, including the material. Implementation of the cheap additives, such as fly ash, helps this method to be competitive on the market. At the same time, the decreasing tool wear helps to avoid the problems with the accuracy of the machined parts [34].

The approach to applying more resistant materials has an advantage, however it is not only possible to increase the wear of tools. Another possibility to improve tool parameters and process performance is proper treatment, such as cryogenic treatment and coating of tool electrodes [34,35,36]. Therefore, further research can also be carried out in this area for developed material and methods of treatment. Future research directions could also include preparation of the materials for different kinds of applications, including applications for different materials, such as advanced ceramics or more precise parts machining [37,38,39]. 

## 4. Conclusions

Five different composition samples (four composites with 2.5, 5.0, 7.5 and 10.0% reinforcement of fly ash in the copper matrix and reference sample without reinforcement) were prepared using the powder metallurgy technique with varying fly ash content and used as an electrode in the EDM process. The SS304 plate was subjected to EDM, and the following findings were made:The rate of material removal increases when the concentration of fly ash in the electrode rises while the other input variables remain the same.Regardless of the use of different composite electrodes during the machining process, the MRR dramatically increased with the increase in current amplitude from 5A to 15A.Tool wear increases as the fly ash content increases during both the 5A and 15A machining scenarios.Increasing the fly-ash content on the tool and current amplitude results in an increase in the surface roughness of the machined surface.

## Figures and Tables

**Figure 1 materials-15-06735-f001:**
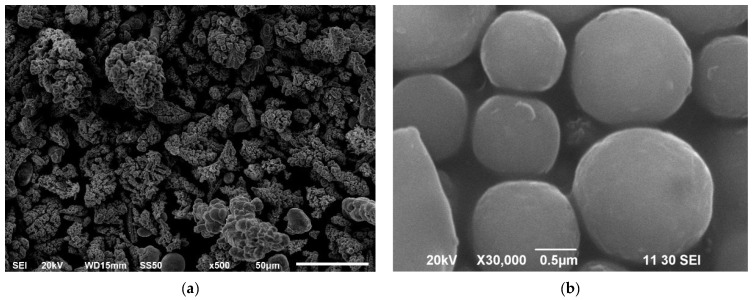
Morphology of matrix and reinforcement particles: (**a**) morphology of copper powder; (**b**) morphology of fly ash particles.

**Figure 2 materials-15-06735-f002:**
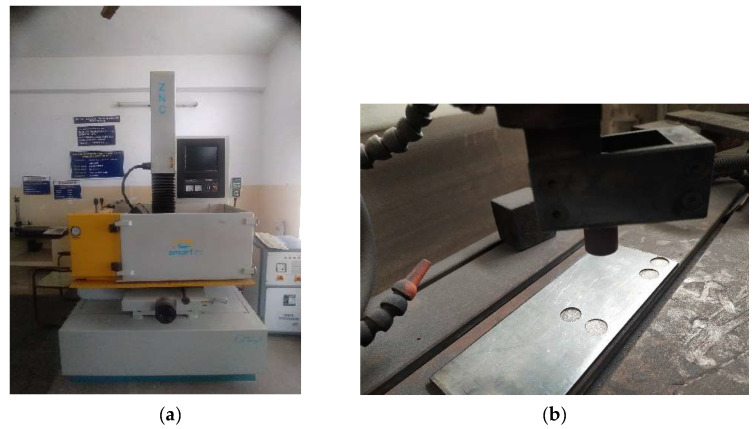
Experimental Setup. (**a**) EDM Machine; (**b**) Tool with workpiece on EDM Machine.

**Figure 3 materials-15-06735-f003:**
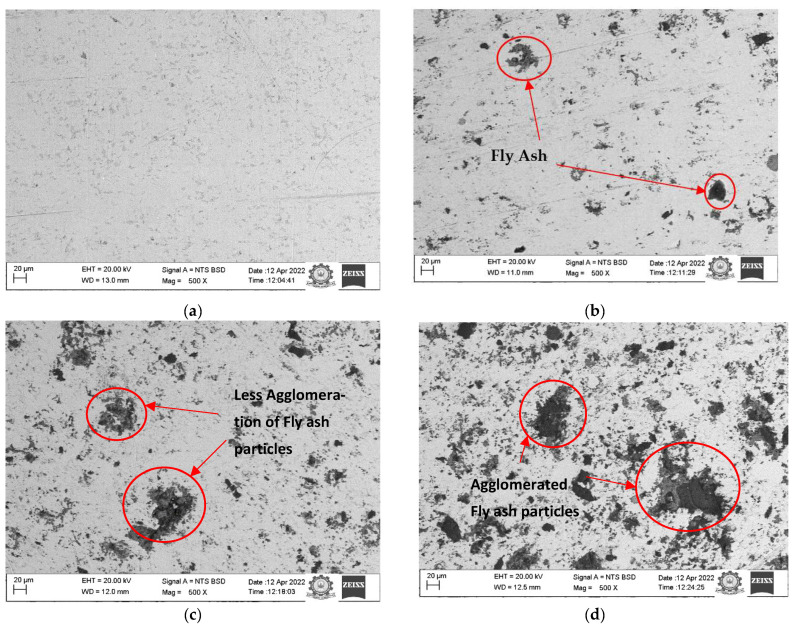
SEM of copper with different proportions of fly ash: (**a**) copper without reinforcement; (**b**) copper with 2.5% fly ash; (**c**) copper with 5.0% fly ash; (**d**) copper with 7.5% fly ash; (**e**) copper with 10.0% fly ash.

**Figure 4 materials-15-06735-f004:**
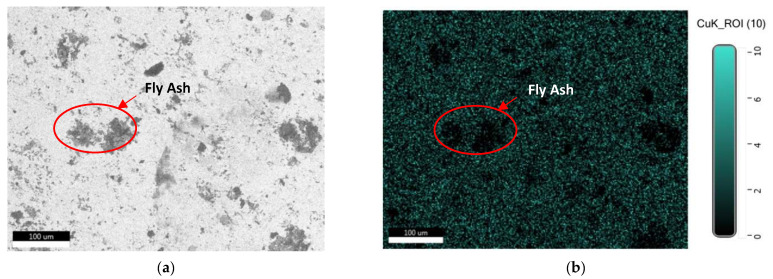
Elements in the composite’s EDX mapping: (**a**) SEM image; (**b**) copper mapping; (**c**) silica mapping; (**d**) alumina mapping; (**e**) oxygen mapping; (**f**) iron mapping.

**Figure 5 materials-15-06735-f005:**
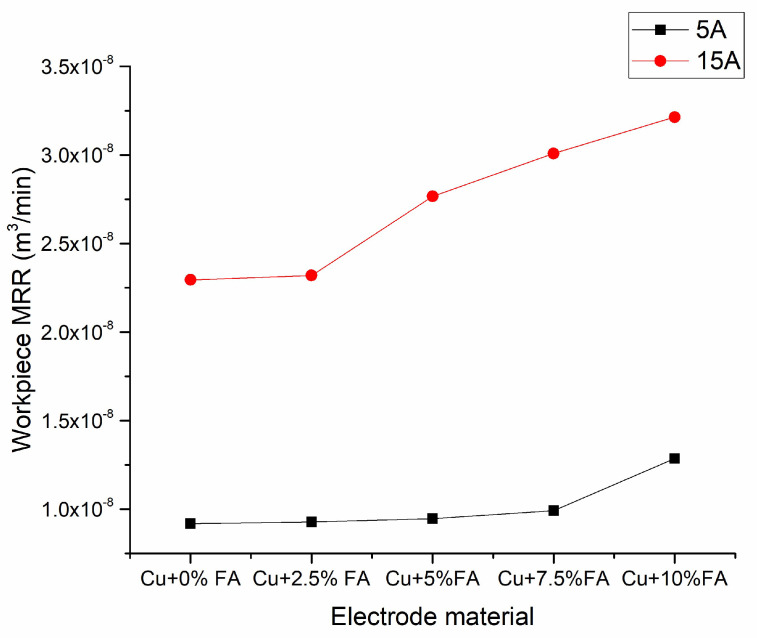
Effect of electrode material and current on MRR.

**Figure 6 materials-15-06735-f006:**
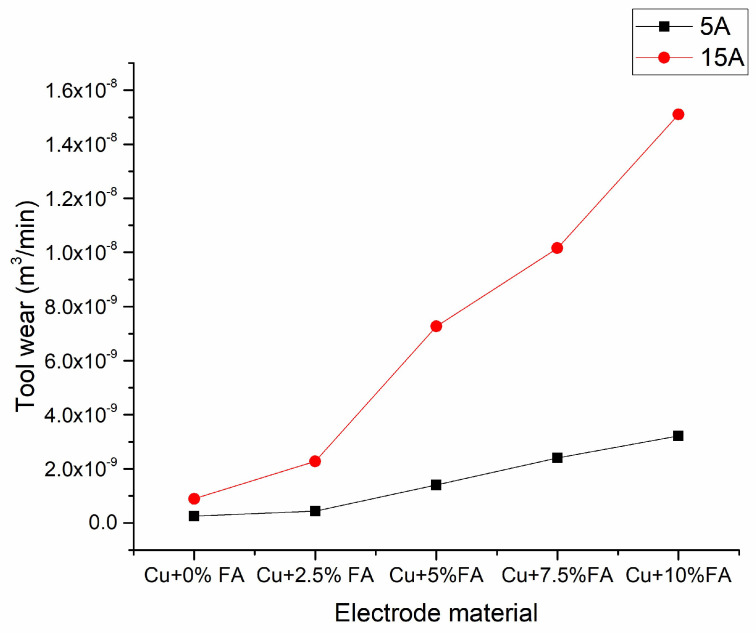
Effect of electrode material and current on tool wear.

**Figure 7 materials-15-06735-f007:**
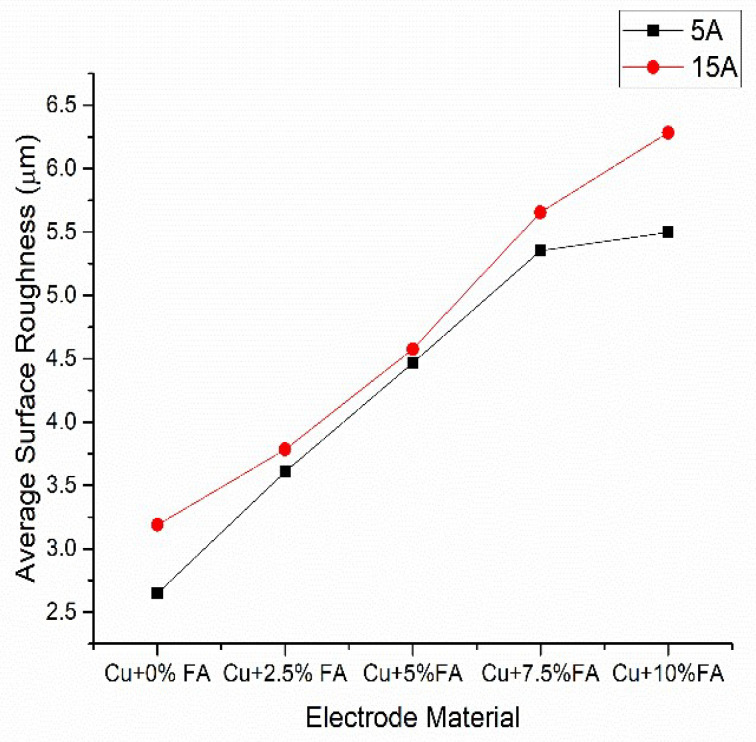
Effect of the electrode material and the average roughness of the surface.

**Table 1 materials-15-06735-t001:** The density of raw material and material after sintering.

Material	Raw Material Density (kg/m^3^)	Density after Sintering (kg/m^3^)
Cu + 2.5% Fly ash	6520	6725
Cu + 5% Fly ash	5576	5794
Cu + 7.5% Fly ash	5147	5244
Cu + 10% Fly ash	4683	4711

**Table 2 materials-15-06735-t002:** Parameters considered during machining.

Parameters	Details
Pulse on time [µs]	100
Pulse off time [µs]	50
Machining depth [mm]	2
Amplitude	5A, 15A

## Data Availability

Not applicable.

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
