# Peer review of "Electric Discharge Machining on Stainless Steel Using a Blend of Copper and Fly Ash as the Electrode Material"

_materials, 2022, doi:10.3390/ma15196735_

Round 1

Reviewer 1 Report

Comments and Suggestions for Authors

Dear authors, The manuscript is well-written and consists of academic novelties with a different approach to the Electric discharge machining on stainless steel using a blend of 2 copper and fly ash as the electrode material. On the other hand, I have some questions and suggestions below.

1. There are some grammatical errors in the manuscript. Please correct all of them.  2. The literature should support the material and method part. For example, there is a lack of literatüre at EDM Responses and experimental measurements 3. Introduction should be reinforced with up-to-date sources. For example, the following publications can be used  • Experimental Investigation of Electro Erosion Machining Parameters of Boron Alloy Steels • Ti6Al4V Alaşımının Elektro Erozyon ile Ä°ÅŸlemesinde Elektrolitik Cu ve CuBe Takım Elektrotlarının Performansının KarşılaÅŸtırılması • Experimental Investigation on Wire Electric Discharge Machining of Biodegradable AZ91 Mg Alloy 4. SEM images should be examined better. What should we understand about the figures, what happened, should be explained by specifying on the figure. 5. Experiment Setup and Procedure, Experiment Plan, Tool Materials, Workpiece Materials, material and method should also be explained better. 6. Technical words should be in accordance with the literature, for example, pulse of time instead of pause time.

Author Response

We would like to thank the reviewer for review and suggestions for the improvement of the manuscript. We have made the changes as suggested by the reviewer. The following changes were applied:

  1. There are some grammatical errors in the manuscript. Please correct all of them.

As per the direction all the grammatical errors are corrected in the revised manuscript

  1. The literature should support the material and method part. For example, there is a lack of literatüre at EDM Responses and experimental measurements

Supporting literatures are added in the revised manuscript

  1. Introduction should be reinforced with up-to-date sources. For example, the following publications can be used • Experimental Investigation of Electro Erosion Machining Parameters of Boron Alloy Steels • Ti6Al4V Alaşımının Elektro Erozyon ile Ä°ÅŸlemesinde Elektrolitik Cu ve CuBe Takım Elektrotlarının Performansının KarşılaÅŸtırılması • Experimental Investigation on Wire Electric Discharge Machining of Biodegradable AZ91 Mg Alloy

Manuscript is enriched with up-to-date resources as per the direction of reviewer.

  1. SEM images should be examined better. What should we understand about the figures, what happened, should be explained by specifying on the figure.

The detailed metallurgical explanation is provided in the discussion part of the revised manuscript and also witnessed with EDAX mapping.

  1. Experiment Setup and Procedure, Experiment Plan, Tool Materials, Workpiece Materials, material and method should also be explained better.

Each part has taken into account thoroughly and detailed explanation is provided in the revised manuscript under materials and methods section.

  1. Technical words should be in accordance with the literature, for example, pulse of time instead of pause time.

As per the direction, the technical words are corrected in accordance with the literature.

Reviewer 2 Report

The paper deals with the Electric discharge machining on stainless steel using a blend of copper and fly ash as the electrode material.
According to the reviewer, the paper is worth publishing at Materials Journal,
but some corrections are needed and then the paper can be accepted for publication in the journal.
While the authors have made considerable research effort,
the presentation of the paper and the results must be proved.
Additionally make the following corrections to the manuscript:

Comment 1
Line 86
The authors must give more details for the equipment (SEM: type, model).

Comment 2
Line 92
The authors must give more details for the equipment (digital balance: type, model).

Comment 3
Line 106
Table 1. displays the green density and sintered density of the produced samples.
The authors should delete the .

Comment 4
Table 1
The authors must insert the units for Raw material density and Density after sintering.

Comment 5
Line 109
The authors must insert a Figure with the EDM Machine with the workpiece and the typical electrode.
The authors must give more details for the EDM Machine (type, model).

Comment 6
Lines 92 and 118
The authors must explain:
Line 92: 0.0001g accuracy
Line 118: accuracy of.001g

Comment 7
Lines 130 and 140
cooper mapping;
Extended text editing 

Comment 8
Line 155
The authors must give more details for the equipment (roughness: type, model).
The authors must give more details for the Average Surface Roughness (μm) (Ra or Rz or.....).

Comment 9
The authors must give more results for taper angle.

Comment 10
The literature study must be enriched. In this respect, authors must read and refer to the following papers:
(a) https://doi.org/10.1007/s12008-022-00859-4 (b)  DOI:10.1504/IJMMM.2016.077712

Author Response

We would like to thank the reviewer for review and suggestions for the improvement of the manuscript. We have made the changes as suggested by the reviewer. The following changes were applied:

Comment 1

Line 86

The authors must give more details for the equipment (SEM: type, model).

The information is added in the revised manuscript

Comment 2

Line 92

The authors must give more details for the equipment (digital balance: type, model).

The information is added in the revised manuscript

Comment 3

Line 106

Table 1. displays the green density and sintered density of the produced samples.

The authors should delete the .

Changes carried out in the revised manuscript

Comment 4

Table 1

The authors must insert the units for Raw material density and Density after sintering.

Units added in the revised manuscript

Comment 5

Line 109

The authors must insert a Figure with the EDM Machine with the workpiece and the typical electrode.

The authors must give more details for the EDM Machine (type, model).

As per the direction the photo of EDM machine and Tool with workpiece is shown figure 2a and 2b

Comment 6

Lines 92 and 118

The authors must explain:

Line 92: 0.0001g accuracy

Line 118: accuracy of.001g

Sorry for the mistake. Its typographical error, the accuracy of the machine is 0.0001g. The same is updated in the revised manuscript.

Comment 7

Lines 130 and 140

cooper mapping;

Extended text editing

The typographical error is rectified in the revised manuscript.

Comment 8

Line 155

The authors must give more details for the equipment (roughness: type, model).

The authors must give more details for the Average Surface Roughness (μm) (Ra or Rz or.....).

The Authors included the details of surface roughness tester. the average roughness is calculated based on Ra and the same is indicated in the revised manuscript.

Comment 9

The authors must give more results for taper angle.

Since it is not a through hole and the measurement of taper is not viable and hence it is not reported.

Comment 10

The literature study must be enriched. In this respect, authors must read and refer to the following papers:

  • https://doi.org/10.1007/s12008-022-00859-4 (b) DOI:10.1504/IJMMM.2016.077712

the Literatures are included in the revised manuscript to enrich the content.

Round 2

Reviewer 1 Report

Thank you very much

Best Regards 

Author Response

Thank you for positive opinion. The article was revised linguistically.

Reviewer 2 Report

Comment 1

Lines 48 and 50

From [15] to [36].

References: References must be numbered in order of appearance in the text (including table captions and figure legends) and listed individually at the end of the manuscript.

The authors must renumber.

Comment 2

Lines 131 - 132

bon (0.07%), Chromium(17.5% to 19.5%), Manganese(2%), Silicon(1%), Phospho- 

rus(0.045%), Sulphur(0.015%), Nickel(8 – 10.5%) and Iron(Remaining %). The machining 

The authors should replace

bon (0.07%), Chromium (17.5% to 19.5%), Manganese (2%), Silicon (1%), Phospho- 

rus (0.045%), Sulphur (0.015%), Nickel (8 – 10.5%) and Iron (Remaining %). The machining 

Line 152

tester(Make: Mitutoyo,

The authors should replace

tester (Make: Mitutoyo,

Line 174

SEM picture(Figure 4a)

The authors should replace

SEM picture (Figure 4a)

Author Response

Thank you for positive opinion. We corrected all mentioned points, including literature and pointed problems in lines 131 - 132, 152, and 174. The article was revised linguistically.